# Machine Learning Applied to Edge Computing and Wearable Devices for Healthcare: Systematic Mapping of the Literature

**DOI:** 10.3390/s24196322

**Published:** 2024-09-29

**Authors:** Carlos Vinicius Fernandes Pereira, Edvard Martins de Oliveira, Adler Diniz de Souza

**Affiliations:** Federal University of Itajubá, Professor José Rodrigues Seabra Campus, Itajubá 37500-903, Brazil; edvard@unifei.edu.br (E.M.d.O.); adlerdiniz@unifei.edu.br (A.D.d.S.)

**Keywords:** systematic mapping, machine learning, edge computing, wearable devices, healthcare

## Abstract

The integration of machine learning (ML) with edge computing and wearable devices is rapidly advancing healthcare applications. This study systematically maps the literature in this emerging field, analyzing 171 studies and focusing on 28 key articles after rigorous selection. The research explores the key concepts, techniques, and architectures used in healthcare applications involving ML, edge computing, and wearable devices. The analysis reveals a significant increase in research over the past six years, particularly in the last three years, covering applications such as fall detection, cardiovascular monitoring, and disease prediction. The findings highlight a strong focus on neural network models, especially Convolutional Neural Networks (CNNs) and Long Short-Term Memory Networks (LSTMs), and diverse edge computing platforms like Raspberry Pi and smartphones. Despite the diversity in approaches, the field is still nascent, indicating considerable opportunities for future research. The study emphasizes the need for standardized architectures and the further exploration of both hardware and software to enhance the effectiveness of ML-driven healthcare solutions. The authors conclude by identifying potential research directions that could contribute to continued innovation in healthcare technologies.

## 1. Introduction

As noted by Farahani et al., the Internet of Things (IoT) has rapidly evolved from a novelty to a significant thematic area with transformative potential across fields such as engineering, monitoring, environmental studies, healthcare, and medicine [1]. The healthcare industry, in particular, has shown a keen interest in IoT and artificial intelligence (AI) due to their potential to enhance efficiency and improve patient outcomes. Furthermore, as Imran et al. highlight, big data analytics can further realize the potential of healthcare systems by transforming heterogeneous healthcare data into actionable knowledge [2]. Consequently, cutting-edge healthcare systems are increasingly incorporating AI-based diagnostics and data-driven treatments to enable early detection and intervention strategies.

However, the time-sensitive nature of healthcare applications necessitates rapid responses. As Yu et al. emphasize in [3], conventional cloud computing-based services are ill-suited for these demands, as computation processes must be uploaded to the cloud and limited bandwidth and network resources are consumed by massive data transmissions. Moreover, the geographical distance between cloud servers and end users can introduce significant latency, which is unacceptable in healthcare applications.

Given these challenges, the post-cloud era, as predicted by Shi et al., mandates that data be processed closer to IoT devices rather than being sent to distant cloud servers [4]. Edge computing, as defined in the same article, offers a solution to this time-sensitivity issue, once it refers to the enabling technologies that allow computation to be performed at the network’s edge, on downstream data on behalf of cloud services and upstream data on behalf of IoT services. In this context, “edge” encompasses any computing and network resources along the path between data sources and cloud data centers.

In alignment with this, Hazra et al. in [5] suggest that the introduction of local edge devices in the healthcare framework can alleviate network congestion and reduce the load on centralized cloud servers.

Considering these developments, this paper proposes a systematic mapping of machine learning-based applications and architectures implemented on edge computing with wearable data in healthcare. The primary objective is to comprehensively understand current trends in this field, identify emerging patterns, and pinpoint gaps that warrant further research and development.

This paper is organized into six main sections. Section 2 describes the research methodology used to collect and compare the different articles and research findings, as well as the criteria for their selection. Section 3 presents the main objectives of the study. Section 4 discusses the results. In Section 5, the research questions are answered. Finally, Section 6 provides a conclusion and highlights future research opportunities.

## 2. Methodology

The objective of this systematic literature mapping is to explore and identify the key insights related to the application of machine learning in healthcare wearable devices. Systematic mapping enables the identification, categorization, and classification of the current state-of-the-art research in the field. Data from primary studies will be systematically organized and analyzed. The methodology employed in this paper follows the framework established by Petersen et al., encompassing research questions, a search protocol, study selection, quality assessment, and data extraction [6].

Furthermore, the PICOC (Population, Intervention, Comparison, Outcome, and Context) protocol was used to guide the formulation of research questions and the development of search strings for filtering relevant articles. To ensure the effective implementation of this protocol, the authors used the Parsifal [7] platform to conduct systematic literature mapping. Through this platform, articles were processed and selected based on the criteria outlined in the following sections.

## 3. Objective

This study aims to **analyze** experience reports and scientific publications through a systematic mapping approach, **with the objective** of identifying solutions and emerging trends in the application of machine learning in healthcare, **in conjunction** with edge computing and wearable devices. This analysis is conducted from the **perspective** of researchers in both the academic and industrial contexts.

### 3.1. Research Questions

In this systematic mapping of literature, we define the main research questions as follows:RQ1: What are the main usages of machine learning and AI in edge computing environments combined with wearable devices for healthcare applications?RQ2: How are machine learning and AI algorithms applied in edge computing environments using wearable devices?RQ3: What architectures (edge layer definitions, wearables used, and machine learning algorithms) are used in modern healthcare applications?

The primary objective of this study, pursued through the research questions, is to understand the key concepts and techniques utilized in healthcare applications involving ML, edge computing, and wearable devices. The first research question serves as the starting point for gathering information from the articles. It focuses on identifying the various applications of ML in this context (QP1). The second research question delves into the methodologies used, exploring how these applications are implemented (QP2). The third research question examines whether there is any established definition or common consensus regarding the architecture of investigated systems (QP3).

### 3.2. Search Protocol and Selection

The authors employed a systematic and detailed approach to extract meaningful and relevant information from the analyzed data. The initial step in identifying articles related to the studied topic involved establishing a well-defined search protocol. The careful selection of keywords played a crucial role in obtaining relevant results. Given that the specific techniques and applications were not fully known, the search protocol encompassed the use of any machine learning technique applied in healthcare contexts. Guided by the research question, the following search string was formulated:


*(“edge computing”) AND (“healthcare”) AND (“wearable”) AND (“Machine Learning” OR “AI” OR “decision making”)*


The search was conducted using Scopus [8] and IEEE [9] databases, with the “TITLE-ABS-KEY” operator added at the start of the search protocol to filter the results based on the presence of keywords in the titles, abstracts, or keywords sections of the articles. This approach ensured that only articles relevant to the specified keywords were retrieved.

The search yielded 70 articles from Scopus and 101 articles from IEEE. As Scopus indexes some IEEE content, there was some overlap between the two databases, resulting in 27 duplicate articles, which were identified and removed.

After removing the duplicates, inclusion and exclusion criteria were applied to further refine the selection. These criteria are detailed in Table 1 and Table 2. Some articles contained the searched keywords only as examples or potential applications, without being central to the study’s focus. Articles that did not comprehensively cover the study’s themes were excluded on this basis.

The exclusion criteria also extended to certain types of publications, such as surveys, keynotes, and non-research articles, as they were not relevant to the study’s scope. Additionally, any articles that were not fully accessible were excluded.

To apply these criteria, a quick review of each article was conducted to assess whether it met the defined conditions. The selection process was binary—if an article exhibited any characteristic matching the exclusion criteria, it was immediately removed from consideration. This process resulted in a final selection of 54 relevant articles. Further selection criteria are discussed next.

Following this step, a quality assessment was conducted on the 54 selected articles, using the following set of questions:Is the research design clearly described and appropriate for the study objectives?Does the study specify the use of edge computing, wearable technology, and ML/AI in healthcare?Are the data collection methods clearly described and appropriate?Is the sample size adequate for the study’s objectives?Does the study provide detailed information on the implementation of edge computing with wearable devices?Are the ML/AI algorithms and models used in the study clearly described?Are the limitations of the study discussed?Are the results consistent and replicable?Are the improvements in healthcare outcomes due to ML/AI integration clearly demonstrated?Does the study contribute new insights or advancements in the field of edge computing, wearable devices, and ML/AI in healthcare?

A scoring system was applied to each question, assigning 0 for “No”, 0.25 for “Not Clear”, 0.5 for “Partially”, and 1 for “Yes”. Articles with a total score above 7 were considered for further analysis. In cases where specific questions were deemed “non-applicable” (e.g., data collection methods in theoretical proposals), the scoring was adjusted accordingly. For these articles, the final score was calculated based on the remaining applicable questions, while ensuring the same proportional threshold of 7/10 was maintained, ensuring fairness across all types of studies.

After this quality assessment, 28 articles were ultimately selected for in-depth analysis in this study.

Figure 1 provides a visual summary of the authors’ approach for the selection, following the PRISMA (Preferred Reporting Items for Systematic Reviews and Meta-Analyses) flow diagram.

### 3.3. Data Extraction and Quantitative and Qualitative Analysis

After selecting the twenty-eight articles that will constitute this work, data extraction was carried out to answer the previously elaborated research questions. The main extracted information was as follows:Title;Authors;Year of publication;Journal/conference;Edge computing details;Wearable devices details;Dataset used;ML/AI algorithms applied;Healthcare application;Data collection methods;Performance metrics;Key findings.

Selected articles were arranged in Table 3. Answers to the research questions were generated, and a discussion and conclusion of the found results were elaborated.

## 4. Results

This section discusses the main results found after analyzing the selected articles presented in Section 3. An initial analysis identified the publication year of each article. As shown in Figure 2, all selected articles are from the last seven years, with the majority published within the last four years.

Another analysis identified which were the main applications obtained from the applied techniques. Figure 3 illustrates the mapping of the healthcare applications found among the selected studies. Fall detection represents more than 32% of the analyzed studies, followed by cardiovascular health monitoring, epileptic seizure detection, heart anomaly detection, and heart disease prediction. Other applications such as food intake monitoring, stroke prediction, blood pressure estimation, and others cover the remaining selected articles. It is important to understand that Yazici et al. in [20] discussed more than one application, so it appears twice on the mapping.

### 4.1. Fall Detection

As stated by Paramasivam et al., falls are accidental events caused by a loss of center of gravity resulting from either a lack of an active effort or an insufficient effort to restore balance [10]. Various factors can cause falls, such as imbalance, poor posture, vision impairment, foot problems, muscle weakness, and others. Supporting this, Ghosh et al. highlighted that almost 40% of injury-related deaths among elderly citizens result from falls [33]. Given this context, it is evident why this is the most extensively researched topic in the field.

A significant challenge in detecting falls lies in distinguishing between a fall and a non-harmful common event, such as a user simply lying down. Additionally, the necessity to alert caregivers as quickly as possible is paramount.

In the domain of wearable devices, there is no standard hardware configuration among the articles; each study employs different hardware, although almost all incorporate an accelerometer as a key component. Table 4 details the hardware utilized in each study.

In the edge layer, Raspberry Pi boards were mentioned in most articles, with an STM board also appearing as the option used by Campanella et al. in [30]. In contrast, Ghosh et al. in [33] utilized the user’s smartphone for this purpose, while Baktir et al. in [18] conducts simulations of the edge layer on a computer. Notably, Chetcuti et al. in [11] does not specify the hardware used for the edge layer. Detailed information on edge layer implementation can be found in Table 5.

Among machine learning (ML) algorithms, no definitive conclusion can be drawn regarding which is the most effective. However, neural networks are frequently utilized. Convolutional Neural Networks (CNNs) are considered the optimal choice by Paramasivam et al. in [10] (where they were combined with a Long Short-Term Memory Network (LSTM)) and Yazici et al. in [20], while isolated LSTMs were selected by Utsha et al. in [26] and Queralta et al. in [31]. The Feedforward Neural Network (FFNN) was employed in Campanella et al. [30].

Other notable approaches include the combination of Federated Learning with a Hidden Markov Model (HMM) and an LSTM by Ghosh et al. in [33], primarily to ensure personal data security. Additionally, XGBoost was applied instead of neural network algorithms by Chetcuti et al. in [11], and a Support Vector Machine (SVM) was used by Baktir et al. in [18]. The Deep Gated Recurrent Unit (DGRU) was utilized by Al-Rakhami et al. in [23].

Regarding training data, no consistent pattern was discernible. Some datasets were used in multiple studies, such as the MobiAct dataset, which was utilized by Queralta et al. in [31] and Ghosh et al. in [33], and the Sisfall dataset, which was employed by Sarabia-Jacome et al. in [26] and Campanella et al. in [30]. Another common practice was the use of data collected by the researchers themselves. In some studies, this was the sole data source, while in others, such as Al-Rakhami et al. in [23], Campanella et al. in [30], and Ghosh et al. in [33], it was combined with external datasets.

The information about the used datasets and the ML algorithms is presented in Table 6. Since all studies used accuracy as a metric, this information is also included.

Accuracy is a commonly used metric for assessing machine learning models, especially in detection tasks. It quantifies the proportion of correctly classified instances, including both true positives and true negatives, relative to the total number of instances.

### 4.2. Cardiovascular Health Monitoring

In contrast to the majority of the found articles, Utsha et al. in [16] and Talha et al. in [25] are monitoring applications focused on the early detection of cardiac diseases and providing a platform for both patient and doctor to monitor the cardiac health. Utsha et al. asserts that detecting diseases at earlier stages improves treatment outcomes [16]. The continuous analysis of vital signs (heart rate, respiratory rate, oxygen saturation, and blood pressure) can predict or detect neonatal pathophysiology, offering the potential to improve outcomes and mitigate neonatal diseases using big data analytics.

In the wearable hardware domain, Utsha et al. in [16] relies on ECG electrodes connected to an ECG module and a microcontroller board, while Talha et al. in [25] does not specify any hardware but suggests the use of biosensors to measure body temperature, heart rate, blood pressure, respiration rate, and blood oxygen saturation.

For the edge layer implementation, Utsha et al. in [16] uses the user’s smartphone to perform the necessary computations, whereas Talha et al. in [25] lacks specific definitions regarding the edge layer implementation. Table 7 shows the results found on the articles.

In the area of machine learning algorithms, both articles tested various models to find the best fit. Utsha et al. in [16] evaluated Convolutional Neural Networks (CNNs), Artificial Neural Networks (ANNs), and Long Short-Term Memory Networks (LSTM), while Talha et al. in [25] assessed Logistic Regression, Naive Bayes (NB), K-Nearest Neighbors (KNN), Support Vector Machine (SVM), Decision Trees (DT), and Random Forest (RF).

Both articles relied on datasets, with Utsha et al. in [16] utilizing the MIT-BIH Arrhythmia dataset and Talha et al. in [25] employing the MIMIC-III dataset.

Additionally, both articles used accuracy as a metric. Utsha et al. in [16] found that the LSTM model provided the best results, achieving an accuracy of 95.94%, while Talha et al. in [25] identified RF as the best option, reaching an accuracy of 95%. Table 8 presents the results for each article.

### 4.3. Epileptic Seizures Detection

Baghersalimi et al. stated that epilepsy is a prevalent neurological disorder affecting approximately 65 million individuals of all ages worldwide [19]. This condition manifests in various forms, with severity ranging from mild to severe, and encompasses a spectrum of seizure types, each with distinct consequences. Consequently, individuals with epilepsy and their families encounter a wide array of challenges and experiences specific to their condition. Among these challenges are issues such as access to quality healthcare, adequate information and coordination of services, and societal stigma. A particularly grave concern is SUDEP (sudden unexpected death in epilepsy), a rare but potentially fatal occurrence that typically transpires during or after a seizure, leading to unexpected deaths within the epilepsy community.

Ingolfsson et al. also notes that the continuous monitoring of brain activity is essential for personalizing patient treatments, which can be performed using electroencephalography (EEG) techniques [12]. Given this context, the necessity for a wearable and fast-response solution becomes apparent.

In the context of wearable devices, both articles focus on EEG systems due to the nature of the detection required, and both are based on hardware proposed in other studies. Despite these similarities, each study proposes different hardware: Ingolfsson et al. in [12] based the application on a brain–computer interface wearable called BioWolf, proposed by Kartsch et al. in [38], while Baghersalimi et al. in [19] suggests the use of e-Glass, a wearable device with four electrodes developed by Sopic et al. in [39].

For the edge layer, each study adopts a different approach. The BioWolf device, as described by Ingolfsson et al. in [12], includes an integrated microprocessor, making the edge layer entirely self-contained within the wearable device. Conversely, Baghersalimi et al. in [19] tested two different platforms (Kendryte K210 and Raspberry Pi Zero) and ultimately selected the Kendryte K210 microcontroller platform. Table 9 presents these data.

Regarding machine learning algorithms, the studies also diverge. Baghersalimi et al. in [19] employed deep neural networks combined with Federated Learning (FL), whereas Ingolfsson et al. in [12] evaluated several algorithms, including Support Vector Machine (SVM), Random Forest (RF), Extra Trees (ETs), and AdaBoost Classifier.

Both studies utilized pre-existing datasets: Ingolfsson et al. in [12] used the CHB-MIT dataset, while Baghersalimi et al. in [19] used a combination of three datasets: EPILEPSIAE, TUSZ, and MIT-BIH.

The results from both studies appear promising. As showed on Table 10, both achievied an accuracy of 100%. Specifically, Ingolfsson et al. in [12] attained this accuracy with the RF and ET algorithms.

### 4.4. Heart Anomaly Detection

Congenital heart anomalies are defects in the structure of the heart or great vessels. They are usually present at birth but can manifest later in life. These anomalies are classified as cardiovascular diseases and can lead to serious health problems. In this context, Firouzi et al. in [22] and Yazici et al. in [20] explored the application of machine learning (ML) algorithms in edge computing for detecting such anomalies.

Regarding wearable hardware, Firouzi et al. in [22] does not specify any particular microchip or sensor, only emphasizing the necessity of using ECG signals for detection. In contrast, Yazici et al. in [20] is based on the ECG module AD8232.

In terms of the edge layer, each study proposes a unique approach. Firouzi et al. in [22] aims to propose a task offloading strategy, using fog computing nodes as the basis for this goal, but does not specify any hardware configuration. Yazici et al. in [20], on the other hand, utilizes a Raspberry Pi Zero as the edge layer, as presented on Table 11.

Different machine learning algorithms were employed in these studies. Firouzi et al. in [22] focused on Convolutional Neural Networks (CNNs) as the ML algorithm, whereas Yazici et al. in [20] tested three classifiers for heart anomaly detection: Deep Neural Networks (DNNs), Random Forest (RF), and CNN, with RF yielding the best results.

Both studies utilized pre-existing datasets: Firouzi et al. in [22] used the MIT-BIH dataset, and Yazici et al. in [20] used the MHEALTH dataset for heart anomaly detection. Accuracy was used as the metric for evaluating the trained ML models in both studies, with the results presented in Table 12.

### 4.5. Heart Disease Prediction

As stated by Jenifer et al., cardiovascular diseases claim 17.9 million lives globally each year [28]. The lives of individuals affected by sudden heart damage could be saved if such events were predicted before their occurrence. Research in this specific area is highlighted by Chakraborty et al. in [24] and Jenifer et al. in [28].

In the domain of wearable devices, the articles adopted different approaches. Chakraborty et al. in [24] provides fewer details about the wearable implementation but suggests using a combination of a blood pressure sensor, a fasting blood sugar sensor, and a heart rate sensor, recommending commercial brands for the latter two without specifying the device names. Conversely, Jenifer et al. in [28] details the use of a temperature sensor (DS18B20), an accelerometer (ADXL1335), and an unspecified pulse sensor.

For the edge layer, Chakraborty et al. in [24] does not explicitly choose an implementation, merely indicating the use of fog nodes to perform this function. In contrast, Jenifer et al. in [28] employs a Raspberry Pi B+ device equipped with a specific Ethernet interface. Table 13 shows these information.

Regarding machine learning algorithms, both articles tested various options to identify the best choice. Chakraborty et al. in [24] evaluated Naive Bayes (NB), K-Nearest Neighbors (KNN), Support Vector Machine (SVM), Random Forest (RF), and Artificial Neural Network (ANN), while Jenifer et al. in [28] selected NB, Decision Trees (DTs), KNN, and SVM. The algorithms were applied to different datasets: Chakraborty et al. in [24] compiled data from various hospitals and health institutions to generate their dataset, while Jenifer et al. in [28] initially trained the ML model with the Human Gait dataset and validated it with data collected during their research. Both studies used accuracy as a metric, with Chakraborty et al. in [24] finding the best results using RF and Jenifer et al. in [28] achieving the highest accuracy with the DT algorithm. Table 14 summarizes their findings and the accuracies achieved.

### 4.6. Other Applications

Other articles have explored various healthcare applications, each employing different approaches and proposed systems.

In the cardiac domain, Ingolfsson et al. in [34] focused on arrhythmia detection; Odema et al. in [32] focused on myocardial infarction detection, and Petroni et al. in [13] proposed a system for atrial fibrillation detection.

Other applications focus on detection: Pazienza and Monte in [14] presented a system aiming to identify COVID-19/influenza, Gokul et al. in [29] addressed freezing of gait detection, and Nandy et al. in [36] focused on communicable disease detection.

Contrasting these detection systems, two articles proposed estimation systems: Pankaj et al. in [17] focused on heart rate estimation, and Banerjee et al. in [21] focused on blood pressure estimation.

Another application type we found was monitoring systems. Four articles were identified in this category: Rachakonda et al. in [15] introduced a food intake monitoring system to track calorie ingestion; Kasaeyan et al. in [35] introduced a pain-monitoring system; and Jiang et al. in [37] introduced a stress-monitoring system.

Furthermore, Elbagoury et al. in [27] proposed a stroke prediction system.

Given the diverse application objectives and types, a variety of hardware is proposed for wearables. The majority of the articles do not specify the models or devices, but some details are given about the types of sensors useful for the proposed systems. Table 15 shows these definitions.

Petroni et al. in [13], Banerjee et al. in [21], and Ingolfsson et al. in [34] based their studies on ECG sensors. Elbagoury et al. in [27] and Kasaeyan et al. in [35] proposed the usage of a body area network (BAN) composed of EMG, ECG, and galvanic skin response (GSR) sensors. Specifically, Elbagoury et al. in [27] also included an EEG sensor, while Gokul et al. in [29] relied on a group of accelerometers attached to the patient’s body. Within the BAN group, Nandy et al. in [36] does not specify the sensors but states the necessity of including data from the pulse rate, body temperature, and other common health monitoring metrics. Additionally, Rachakonda et al. in [15] proposed the use of glasses equipped with cameras.

In the more specific category, Pankaj et al. in [17] based their work on a photoplethysmogram optical sensor using the GRAVITY SEN0203 device; and Jiang et al. [37] employed a commercial device called RespiBAN, which includes ECG, electrodermal activity (EDA), EMG, respiratory signal, temperature, and accelerometer sensors. Another notable article on this matter is that by Pazienza and Monte in [14], which describes a multi-sensor hardware installed on a 3D-printed mask. Conversely, Ghosh et al. in [33] did not provide any information about the wearables. On Table 16 a better visuallization for these information can be found.

Regarding the edge layer, most articles are specific about the implementation. It is noteworthy that Pazienza and Monte in [14], Elbagoury et al. in [27], and Jiang et al. in [37] proposed using the user’s smartphone as the edge layer, with everything connected to the wearables. Other articles, such as those by Petroni et al. in [13], Pankaj et al. in [17], Gokul et al. in [29], Odema et al. in [32], Ingolfsson et al. in [34], Kasaeyan et al. in [35], and Nandy et al. in [36], focused on commercial microcontrollers. For instance, Gokul et al. in [29] tested two different boards and identified the Raspberry Pi 3 Model B as the best option. Meanwhile, Rachakonda et al. in [15] and Banerjee et al. in [21] proposed using a “common computer” as the edge layer. All these information are presented on Table 17.

In terms of machine learning algorithms, neural networks are predominantly used. Banerjee et al. in [21], Gokul et al. in [29], Kasaeyan et al. in [35], Nandy et al. in [36], and Jiang et al. in [37] tested at least two different algorithms, with only Kasaeyan et al. in [35] finding two algorithms with the same results. Additionally, Pazienza and Monte in [14] utilized the XGBoost Algorithm, Rachakonda et al. in [15] employed the Single-Shot MultiBox Detector, suitable for the proposed application. Ingolfsson et al. in [34] adopted a Temporal Convolutional Network (TCN), and Petroni et al. in [13] used the Deep L1-PCA, an algorithm originally presented for brain connectivity measurements, and Odema et al. in [32] proposed a CNN algorithm combined with Early Exit (EEx) and the Neural Architecture Search (NAS) technique. Elbagoury et al. in [27] based the application on two different algorithms: the GMDH Neural Network for stroke prediction and the Sparse Autoencoder for stroke diagnosis.

For data collection, only Rachakonda et al. in [15], Pankaj et al. in [17], and Elbagoury et al. in [27] gathered data through experiments, while the other articles relied on pre-existing datasets.

Since the selected articles cover a wide range of applications, there is no consistent pattern in the use of evaluation metrics. For those studies that report accuracy, it is used as a key metric and is presented in Table 18. For studies that employ different metrics, these are listed in a separate column. Four additional metrics commonly used in the reviewed studies include the following:**Recall**: Measures the completeness of the model’s output, indicating how many relevant instances it correctly identified.**Precision**: Measures the precision of the model’s output, indicating how many of the instances it identified were actually relevant.**F1-score**: Combines precision and recall into a single metric.**Mean absolute error (MAE)**: Measures the average absolute difference between predicted and actual values, providing a simple measure of error magnitude.

It is important to note that these metrics should not be used to directly compare results across different studies, as they target varied applications and objectives (e.g., prediction versus detection tasks). Instead, they are presented to showcase the specific evaluation methods and outcomes reported in each article.

## 5. Discussion

The results and answers to the research questions presented in Section 4 are discussed below.


**RQ1: What are the main usages of machine learning and AI in edge computing environments combined with wearable devices for healthcare applications?**


As shown in Section 4, a variety of applications are being studied that leverage ML algorithms in edge computing combined with wearable devices for healthcare. The most common application is fall detection, particularly for elderly monitoring, representing 31.25% of articles. Additionally, there are several applications focused on processing ECG signals and their various uses. Taking a more holistic view, “detection” applications are the most prevalent, accounting for 68% of the articles in this area.


**RQ2: How are machine learning and AI algorithms applied in edge computing environments using wearable devices?**


There is no consensus on how to apply ML algorithms. However, the combination of public data collection for training and experimental data for validation appears to be an effective approach. The choice of algorithm often depends on the specific application, but neural networks are the most commonly used, likely due to their suitability for classification tasks, which are necessary in almost all cases.


**RQ3: What architectures (edge layer definitions, wearables used, machine learning algorithms) are used in modern healthcare applications?**


As discussed in Section 4, there is significant variation in the architectures employed across the reviewed articles, even within the same application domains. No standardized approach has been established for the choice of wearable devices or the configuration of the edge layer. A common strategy is to leverage the user’s smartphone as the edge layer, along with commercial microcontroller platforms. In these cases, wearable devices typically transmit data via Bluetooth to the smartphone or microcontroller, where data processing occurs. This method is practical as it allows for the offloading of the computationally intensive ML training tasks to more powerful external systems, which helps manage resource constraints such as power consumption and processing capabilities. Figure 4 demonstrates these common architectures, showcasing how smartphones and microcontrollers interact with wearables. In some cases, the wearables themselves are equipped with embedded microcontrollers, enabling on-device processing.

The selection of machine learning algorithms also shows no unified trend. While neural networks are the most frequently used, a wide variety of algorithms and architectures are adopted across different applications. In some cases, multiple algorithms are combined to improve performance, and alternative models like XGBoost and SVM are also employed in specific scenarios. This diversity reflects the specific needs of different healthcare applications and the flexibility in choosing architectures to meet them.

## 6. Conclusions

The primary goal of this systematic review was to map the existing literature on the integration of machine learning (ML), edge computing, and wearable devices in healthcare applications. This review underscores the vast potential and growing importance of these technologies in applications such as fall detection, cardiovascular monitoring, and disease prediction. Despite the frequent use of neural networks—particularly Convolutional Neural Networks (CNNs) and Long Short-Term Memory Networks (LSTMs)—the literature still lacks a thorough discussion of the rationale behind model selection and a comparative analysis of how different models perform relative to one another in various healthcare contexts.

Common challenges identified across the reviewed studies include power consumption, storage limitations, and performance constraints. These concerns are particularly relevant given that many of the applications run on microcontrollers and smartphones, where resource efficiency is critical. Additionally, the preprocessing of data and the removal of noise, especially from Internet of Things (IoT) devices, present another significant challenge. In real-world scenarios, noisy and sparse data can undermine prediction accuracy, further complicating the deployment of reliable ML models in healthcare settings.

To address these gaps, future research should prioritize a comparative evaluation of ML models, focusing on performance trade-offs, computational efficiency, and real-time applicability. Moreover, detailed insights into the rationale for model selection and performance benchmarking would be invaluable in guiding the development of more optimized and standardized architectures. Such advancements would accelerate the adoption of ML-driven healthcare solutions, ensuring their effectiveness, scalability, and sustainability in a wide range of healthcare applications.

This research was conducted using a systematic mapping methodology. Initially, 171 articles were refined using a search protocol. After applying inclusion and exclusion criteria and conducting a quality assessment, 28 articles were selected for analysis. These papers provided the basis for this review. This article contributes to the analysis of current studies on healthcare applications that utilize machine learning, edge computing, and wearable devices, offering an overview of the main architectures and applications proposed by researchers worldwide. 

## Figures and Tables

**Figure 1 sensors-24-06322-f001:**
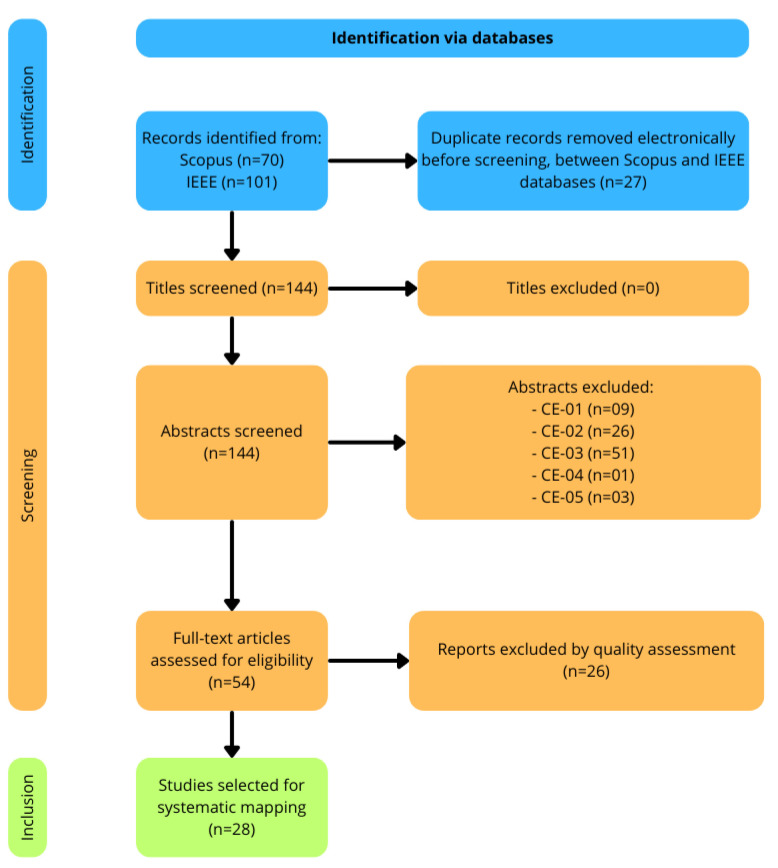
PRISMA flow diagram showing the methodology used for systematic mapping.

**Figure 2 sensors-24-06322-f002:**
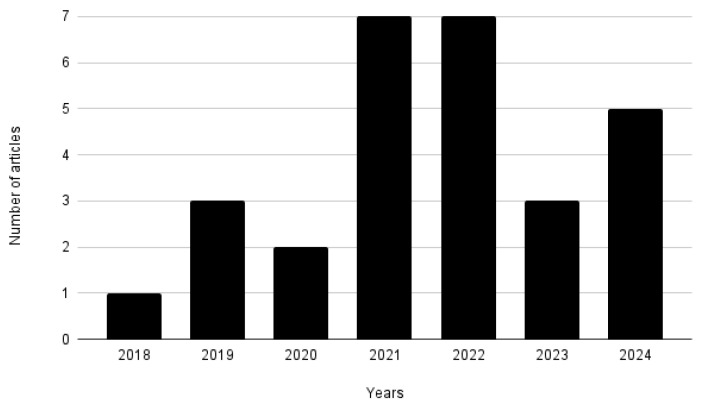
Amount of articles by publication year.

**Figure 3 sensors-24-06322-f003:**
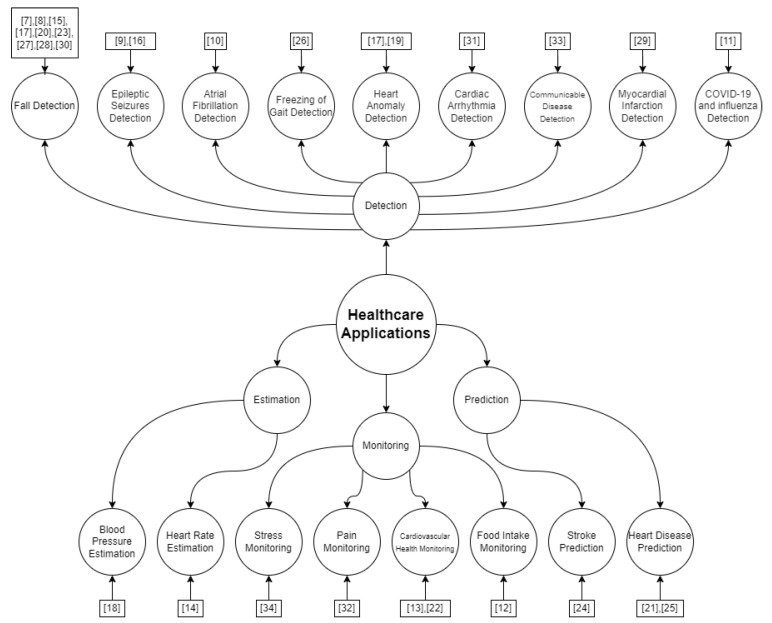
Conceptual mapping of healthcare applications based on the selected articles.

**Figure 4 sensors-24-06322-f004:**
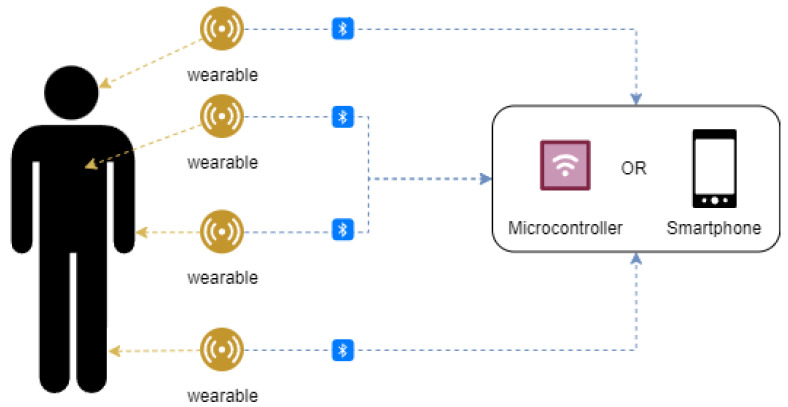
A diagram illustrating the use of commercial microcontrollers and smartphones in conjunction with wearable devices for healthcare applications.

**Table 1 sensors-24-06322-t001:** Set of inclusion criteria.

Criteria	Description
CI-01	Selected articles with a real case application
CI-02	Selected articles with theoretical analysis
CI-03	Only selected articles with all the themes included

**Table 2 sensors-24-06322-t002:** Set of exclusion criteria.

Criteria	Description
CE-01	Does not deal with healthcare
CE-02	Does not involve wearables
CE-03	Publications of workshops, keynote speeches, surveys, and similar works
CE-04	Retracted article
CE-05	Not fully available

**Table 3 sensors-24-06322-t003:** Articles selected through the systematic mapping of the literature.

Nº	Ref	Title	Healthcare Application
1	[10]	Development of artificial intelligence edge computing based wearable device for fall detection and prevention of elderly people	Fall detection
2	[11]	Data Processing Using Edge Computing: A Case Study For The Remote Care Environment	Fall detection
3	[12]	Towards Long-term Non-invasive Monitoring for Epilepsy via Wearable EEG Devices	Epileptic seizure detection
4	[13]	Atrial Fibrillation Detection by Multi-Lead ECG Processing at the Edge	Atrial fibrillation detection
5	[14]	Introducing the Monitoring Equipment Mask Environment	COVID-19 and influenza detection
6	[15]	ILog: An Intelligent Device for Automatic Food Intake Monitoring and Stress Detection in the IoMT	Food intake monitoring
7	[16]	CardioHelp: A smartphone application for beat-by-beat ECG signal analysis for real-time cardiac disease detection using edge-computing AI classifiers	Cardiovascular health monitoring
8	[17]	Edge-Based Computation of Super-Resolution Superlet Spectrograms for Real-Time Estimation of Heart Rate Using an IoMT-Based Reference-Signal-Less PPG Sensor	Heart rate estimation
9	[18]	SDN-Based Multi-Tier Computing and Communication Architecture for Pervasive Healthcare	Fall detection
10	[19]	Decentralized Federated Learning for Epileptic Seizures Detection in Low-Power Wearable Systems	Epileptic seizure detection
11	[20]	A smart e-health framework for monitoring the health of the elderly and disabled	Heart anomaly detection; fall detection
12	[21]	Blood Pressure Estimation from ECG Data Using XGBoost and ANN for Wearable Devices	Blood pressure estimation
13	[22]	Task Offloading for Edge-Fog-Cloud Interplay in the Healthcare Internet of Things (IoT)	Heart anomaly detection
14	[23]	FallDeF5: A Fall Detection Framework Using 5G-Based Deep Gated Recurrent Unit Networks	Fall detection
15	[24]	Real-Time Cloud-Based Patient-Centric Monitoring Using Computational Health Systems	Heart disease prediction
16	[25]	Paving the way to cardiovascular health monitoring using Internet of Medical Things and Edge-AI	Cardiovascular health monitoring
17	[26]	Efficient Deployment of Predictive Analytics in Edge Gateways: Fall Detection Scenario	Fall detection
18	[27]	Mobile AI Stroke Health App: A Novel Mobile Intelligent Edge Computing Engine based on Deep Learning models for Stroke Prediction – Research and Industry Perspective	Stroke prediction
19	[28]	Edge-based Heart Disease Prediction Device using Internet of Things	Heart disease prediction
20	[29]	Gait Recovery System for Parkinson’s Disease using Machine Learning on Embedded Platforms	Freezing of gait detection
21	[30]	A Novel Embedded Deep Learning Wearable Sensor for Fall Detection	Fall detection
22	[31]	Edge-AI in LoRa-based Health Monitoring: Fall Detection System with Fog Computing and LSTM Recurrent Neural Networks	Fall detection
23	[32]	EExNAS: Early-Exit Neural Architecture Search Solutions for Low-Power Wearable Devices	Myocardial infarction detection
24	[33]	FEEL: FEderated LEarning Framework for ELderly Healthcare Using Edge-IoMT	Fall detection
25	[34]	ECG-TCN: Wearable Cardiac Arrhythmia Detection with a Temporal Convolutional Network	Cardiac arrhythmia detection
26	[35]	An Edge-Assisted and Smart System for Real-Time Pain Monitoring	Pain monitoring
27	[36]	Analysis of Communicable Disease Symptoms Using Bag-of-Neural Network at Edge Networks	Communicable disease detection
28	[37]	A Resilient and Hierarchical IoT-Based Solution for Stress Monitoring in Everyday Settings	Stress monitoring

**Table 4 sensors-24-06322-t004:** Fall detection: wearable hardware.

Ref	Hardware Type	Chosen Hardware
[10]	Accelerometer	MPU6050
[11]	Commercial device	Zephyr BioHarness
[18]	Inertial measurement unit	EXEL ExLs3
[20]	Accelerometer	MPU-9250
[23]	Smartwatch	Not specified
[26]	Accelerometer	ADXL345
[30]	Accelerometer and pressure sensor	LSM6DSOX & LPS25HB
[31]	Accelerometer	MPU-9250
[33]	Accelerometer	ADXL345

**Table 5 sensors-24-06322-t005:** Fall detection: edge layer implementation.

Ref	Edge Layer Implementation
[10]	Raspberry PI 3 Model B
[11]	Not specified
[18]	Simulated only
[20]	Raspberry Pi
[23]	Raspberry Pi
[26]	Raspberry Pi 2 model B
[30]	STM32U5 microcontroller
[31]	Raspberry Pi 3 Model B
[33]	Smartphone

**Table 6 sensors-24-06322-t006:** Fall detection: datasets, best models and their accuracies.

Ref	Dataset	Best Model	Accuracy
[10]	Self-collected data	CNN-LSTM	97.00%
[11]	Self-collected data	XGBoost	100.00%
[18]	Self-collected data	SVM	93.80%
[20]	DMLSmartActions	CNN	92.86%
[23]	Smartwatch, Smartfall	DGRU	96.40%
[26]	Sisfall	LSTM	98.40%
[30]	Sisfall, self-collected data	FFNN	99.38%
[31]	MobiAct	LSTM	95.30%
[33]	MobiAct, self-collected data	HMM-LSTM	92.16%

**Table 7 sensors-24-06322-t007:** Cardiovascular health monitoring: wearable and edge hardware.

Ref	Wearable Hardware	Edge Hardware
[16]	AD8232 ECG module	Smartphone
[25]	Bio-sensor for vital signals	Not specified

**Table 8 sensors-24-06322-t008:** Cardiovascular health monitoring: datasets, best models, and their accuracies.

Ref	Dataset	Best Model	Accuracy
[16]	MIT-BIH Arrhythmia	LSTM	95.94%
[25]	MIMIC-III	RF	95.00%

**Table 9 sensors-24-06322-t009:** Epileptic seizure detection: wearable and edge hardware.

Ref	Wearable Hardware	Edge Hardware
[12]	BioWolf [38]	BioWolf [38]
[19]	E-Glass [39]	Kendryte K210

**Table 10 sensors-24-06322-t010:** Epileptic seizure detection: datasets, best models and their accuracies.

Ref	Dataset	Best Model	Accuracy
[12]	CHB-MIT	RF and ET	100.00%
[19]	EPILEPSIAE, TUSZ, MIT-BIH	DNN	100.00%

**Table 11 sensors-24-06322-t011:** Heart anomaly detection: wearable and edge hardware.

Ref	Wearable Hardware	Edge Hardware
[22]	Not specified	Not specified
[20]	AD8232 ECG module	Raspberry Pi Zero

**Table 12 sensors-24-06322-t012:** Heart anomaly detection: datasets, best models, and their accuracies.

Ref	Dataset	Best Model	Accuracy
[22]	MIT-BIH	CNN	96.00%
[20]	MHEALTH	RF	99.97%

**Table 13 sensors-24-06322-t013:** Heart disease prediction: wearable and edge hardware.

Ref	Wearable Hardware	Edge Hardware
[24]	Blood pressure, fasting blood sugar, heart rate sensors	Not specified
[28]	DS18B20, ADXL1335, pulse sensor	Raspberry Pi B+

**Table 14 sensors-24-06322-t014:** Heart disease prediction: datasets, best models, and their accuracies.

Ref	Dataset	Best Model	Accuracy
[24]	Self-collected data	RF	97.32%
[28]	Human Gait, self-collected data	DT	94.00%

**Table 15 sensors-24-06322-t015:** System types and applications.

Ref	System Type	Application
[13]	Detection	Atrial fibrillation detection
[14]	Detection	COVID-19 and influenza detection
[15]	Monitoring	Food intake monitoring
[17]	Estimation	Heart rate estimation
[21]	Estimation	Blood pressure estimation
[27]	Prediction	Stroke prediction and diagnosis
[29]	Detection	Freezing of gait detection
[32]	Detection	Myocardial infarction detection
[34]	Detection	Cardiac arrhythmia detection
[35]	Monitoring	Pain monitoring
[36]	Detection	Communicable disease detection
[37]	Monitoring	Stress monitoring

**Table 16 sensors-24-06322-t016:** Other applications: wearable description.

Ref	Hardware Type Proposed	Hardware Specifications
[13]	ECG sensor	Not specified
[14]	Equipment mask	MAX30102,BlueDot TMP117
[15]	Glasses with cameras	Not specified
[17]	PPG sensor	GRAVITY SEN0203
[21]	ECG sensor	Not specified
[27]	EEG, ECG, EMG, and GSR sensors	Not specified
[29]	Accelerometers	Not specified
[32]	not specified	Not specified
[34]	ECG sensor	Not specified
[35]	EMG, ECG, GSR sensors	Not specified
[36]	BAN system	Not specified
[37]	ECG, EDA, EMG, respiratory signal, temperature, and accelerometer	RespiBAN

**Table 17 sensors-24-06322-t017:** Other applications: edge layer description.

Ref	Proposed Edge Layer	Edge Layer Specifications
[13]	Commercial microcontroller	ESP32
[14]	Mobile phone	-
[15]	Computer	Not specified
[17]	Commercial microcontroller	ARM Cortex-A72
[21]	Computer	Intel i7-5500U CPU
[27]	Mobile phone	-
[29]	Commercial microcontroller	ATMega2560 board
[32]	Commercial microcontroller	EFM32 Giant Gecko
[34]	Commercial microcontroller	ARM Cortex M4F
[35]	Commercial microcontroller	A Raspberry Pi 3 Model B
[36]	Commercial microcontroller	Raspberry Pi
[37]	Mobile phone	-

**Table 18 sensors-24-06322-t018:** Other applications: datasets and best models.

Ref	Dataset	Best ML Model	Accuracy	Other Metrics
[13]	DS1 dataset	Deep l1-PCA	-	F1-Score: 78.10%
[14]	Non-public dataset (the data used by Pazienza and Monte in [14] were collected from a study cohort from West Virginia University (WVU) Hospital, but it is not public.)	XGBoost	94.80%	-
[15]	Generated by the researchers	Single-Shot MultiBox Detector	98.00%	-
[17]	IEEE SPC, BAMI, Generated by the researchers	CNN	-	MAE: 1.27 BPM
[21]	MIMIC-II dataset	XGBoost	-	Precision: 75.69%, Recall: 89.11%
[27]	Generated by the studies	Sparse Autoencoder/GMDH Neural Network	96.68%	-
[29]	DAPHNet	ProtoNN	-	Recall: 93.58%
[32]	PTB ECG, w-HAR dataset	EExNAS	98.54%	-
[34]	ECG5000 dataset	TCN	94.20%	-
[35]	BioVid heat pain	SVM and RF	79.00%	-
[36]	Brazil COVID-19 dataset	BoNN	99.80%	-
[37]	WESAD dataset	CNN	98.92%	-

## Data Availability

No new data were created during this study.

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
