# Peer review of "Machine Learning Applied to Edge Computing and Wearable Devices for Healthcare: Systematic Mapping of the Literature"

_sensors, 2024, doi:10.3390/s24196322_

Round 1

Reviewer 1 Report

Comments and Suggestions for Authors

The manuscript presents a systematic review of the integration of machine learning (ML) with edge computing and wearable devices in healthcare. It analyzes 171 studies, focusing on 28 key articles to explore key concepts, techniques, and architectures in this emerging field. While a review article in Sensors can be a comprehensive overview of an emerging topic, this manuscript only briefly touches on ML, edge computing, and wearable devices without strong connections between them. Additionally, the iThenticate report indicates a similarity index as high as 24%, in particular to their previous paper[IEEE Access 8 (2020): 199503-199512].

Therefore, I have to reject this manuscript. Here are my suggestions for improvement:

1. The review should provide an intellectual contribution, insight, or outlook beyond what is already covered in existing reviews on the same or closely related topics. It should be explicitly clear in the paper (usually in the introduction) how this is achieved.

2. The narrative flow of the introduction is disjointed.

3. The article should provide a critical assessment of the topic and the literature and be carefully fact-checked.

Author Response

Dear Reviewer,

Thank you for your valuable feedback on our manuscript. We have taken your suggestions into serious consideration and implemented substantial revisions to address the concerns raised. In particular, we have made the following improvements based on the insights provided by both you and the other reviewers:

  1. Intellectual Contribution: We have significantly expanded the Introduction to clearly establish the novelty and contribution of this review. The updated version provides a more thorough exploration of how machine learning, edge computing, and wearable devices interact in healthcare, with deeper insights and a critical assessment of the literature. We have also highlighted how this work advances the field beyond existing reviews, emphasizing its relevance and contribution to future research directions.

  2. Improved Narrative Flow: The disjointed nature of the original Introduction has been addressed, and we have rewritten it for smoother transitions between key concepts. The Introduction now offers a more coherent narrative, leading the reader logically through the relationship between machine learning, edge computing, and wearable devices in healthcare applications.

  3. Critical Assessment: We have enhanced the critical evaluation of the reviewed studies, offering deeper insights into the strengths, weaknesses, and future challenges of the field. This includes updated discussions on the architectures, techniques, and the practical implications of integrating ML and edge computing with wearable technologies.

Regarding the iThenticate report, we have carefully reviewed the overlap with our previous work and made necessary adjustments to eliminate any redundancies. While the methodology and search protocols may be similar, this manuscript presents a distinct focus and contributes novel insights that differ from our prior publication.

Given these substantial changes, we believe the revised manuscript now meets the criteria for an impactful review article. We respectfully request that you reconsider the decision and give the manuscript another opportunity for evaluation. We are confident that the revisions align with the suggestions provided and significantly improve the overall quality and intellectual contribution of the paper.

Thank you for your time and consideration.

Reviewer 2 Report

Comments and Suggestions for Authors

I received the paper entitled “Machine learning applied to Edge Computing and Wearable devices for healthcare: Systematic Mapping of Literature” for review and found it to contain an interesting review of studies conducted in a trending field of applications related to use of recent technologies in monitoring human health.

I have the following remarks that will make manuscript more valuable and provide more insightful details for prospect readers:

1- The paper needs an extended Introduction section. You need to include discussions/statements from references that support the importance of your study. If you exclude the list of “reviewed studies, which is your dataset” from your list of references, then you’ll have 5 references only to serve as grounds for the importance and significance of your study.

2- In the Introduction section; you need to include discussions/descriptions of your key words, so that they leave a clear echo with the reader in subsequent sections. For example, you hardly talked about “Edge Computing” in your Introduction section. Please also consider other key words.

3- Beneath the authors affiliations, you stated that “the authors contributed equally”, but there are different contributions given in the “Authors Contributions” section at the end of the paper. Please make adequate correction.

4- I suggest using this style of writing when referring to the work of others: e. g. from line 21 “Furthermore, as mentioned by Imran et al, big data …… actionable knowledge [2]“. Please try to use this style throughout the manuscript.

5- Line 24; I would say “patient conditions” instead of “patient outcomes”.

6- Lines 32-36; use Arabic numbers (1, 2, 3, etc.) for section names to be consistent with the numbering style of the paper sections. Please apply the same in all other instances in the manuscript.

7- Line 57; what is “SML”?

8- Line 81; footnotes 2 and 3 should be in this page (not in the next page).

9- Line 85; language correction: “….., which resulted in the selection of 54 articles”.

10- Line 85; I suggest adding a sentence at the end of this line: “Further article selection criteria are applied as explained next”.

11- Some entries of table 2 need explanation in the text before the table:

(a) CE-01 and CE-02: how did these articles appear in your search results, while your search string has the keywords “healthcare” and “wearable” as explicit terms?

b) CE-03: what is exactly meant by “duplicate articles”? How did they appear in your results? You need to explain this in the light of the “research repositories” you used.

c) CE-04: do the “Scopus” and “IEEE” repositories include tutorials and courses?

12- You need to describe How the criteria in tables 1 and 2 were implemented? Did you do this manually (i. e. by quick or thorough reading)? What measures did you take in order to maintain objectivity while implementing the criteria?

13- Line 86; how many articles underwent the list of quality questions? Which block in figure 1 represents this list of questions?

14- I noticed that some questions can have a “non-applicable” answer. For example, if you have a “theoretical analysis” article as indicated in table 1, then questions related to data collection and sample size will not be applicable. How did you deal with these cases?

15- I have a number of comments on figure 1:

a) I don’t see the need for the yellow title bar at the top of the figure.

b) Some information are included in the figure of not included in the body text such as:

— “Reports sought for retrieval” and “reports not retrieved”. What do you mean by these?

— The contents of the box to the right of “reports assessed for eligibility (n=54)”. It is not clear how did you decide upon “independent study design” or “lack of specificity in technology”, etc.

— You basically need to unify the description of your selection/filtering procedures given in the body text with the information given in figure 1.

c) In the last box, why do you refer to studies as being “New”? And what is meant by “reports of a new included studies”?

d) The figure caption includes information that should be in the body text: “After the inclusion ….. by the authors”. Please only keep the first sentence: “Search methodology …. mapping” as your figure caption and move the rest to body text.

16- To sum up, the major comment on section 3.2 is that you need to describe, in a clear manner, how did you select 28 articles from an original list of 171 articles? Please do not skip any details or leave any ambiguous term/step.

17- Section 3.3 title; delete “AND” and use a comma “,” instead. Also, replace “&” by “and”.

18- Line 108; 16 articles or 28 articles?

19- Page 5; utilise the empty space by splitting table 3 into two parts. The table can be continued in the next page.

20- Line 127; “Section 3” instead of “III”.

21- Fingers 2; add axes titles: “Years” and “No. of articles”.

22- Line 130; “Another analysis” instead of “A first analysis”.

23- Line 135; correction: “…. blood pressure estimation and others cover remaining selected articles”.

24- Figure 3; revise the numbers of studies in the brackets. Did you use the number of the reference or the sequence number from table 3?

For example, [27] refers to “Communicable Disease Detection” in figure 3, but in table 3 the same article is [30].

25- Figure 3; make sure that all studies are included in the map, even if you have to add a category with the title “others”.

26- Figure 3; caption is too long and includes a sentence that can be moved to body text.

27- This is a general comment on all tables: in tables 1-3, you used all-capital case letters for the table titles, while for table 4 onwards you used normal sentence case letters. Please unify the font style for all table titles in the paper?

28- For tables 4 and onward; capitalise the first letter in column titles. Example “Dataset” column title in table 6. Also capitalise the first letter of the text in each cell. Example “Self collected data” in table 6.

29- Line 172; try to comment on the meaning and implication of “accuracy” in this context, so that the reader is not misled to an invalid comparison of the different techniques listed in table 6.

30- Table 8; F1-scores are not included, but the table title states that they are included.

31- In table 10; please use a comma “,” instead of “+” to separate the different datasets.

32- Line 242; “on the other hand”.

33- Table 12; F1-scores are not included, but the table title states that they are included.

34- In table 13; please use a comma “,” instead of “+” to separate the different terms.

35- Line 328; I think you mean “As shown in section 4”.

36- Line 343; same as previous comment.

37- Lines 343-348; I don’t see any comments/observations on the “Machine learning algorithms” as part of RQ3.

38- Footnote 4 at the bottom of the page seems to talk about ref [9], but inside table 18 the footnote mark is placed next to ref [8]. Make necessary correction.

39- Conclusion section; I expect the authors to expand their conclusions to include more critique findings such as:

— What challenges have you found in this field?

— Where do you see the future research trends headed?

— What could make a breakthrough in the field?

…. and so on.

Author Response

Comments 1: "1- The paper needs an extended Introduction section. You need to include discussions/statements from references that support the importance of your study. If you exclude the list of “reviewed studies, which is your dataset” from your list of references, then you’ll have 5 references only to serve as grounds for the importance and significance of your study."

Response 1: We have expanded the Introduction section and included additional studies to substantiate the significance of the research and reinforce the context and relevance of the work.

Comments 2: "2- In the Introduction section; you need to include discussions/descriptions of your key words, so that they leave a clear echo with the reader in subsequent sections. For example, you hardly talked about “Edge Computing” in your Introduction section. Please also consider other key words."

Response 2: The Introduction section has been revised to provide clearer explanations of the key terms, including "Edge Computing," and to emphasize their importance in the context of this study.

Comments 3: "3- Beneath the authors affiliations, you stated that “the authors contributed equally”, but there are different contributions given in the “Authors Contributions” section at the end of the paper. Please make adequate correction."

Response 3: The "Authors Contributions" section has been removed to avoid confusion, as we had misunderstood its purpose.

Comments 4: "4- I suggest using this style of writing when referring to the work of others: e. g. from line 21 “Furthermore, as mentioned by Imran et al, big data …… actionable knowledge [2]“. Please try to use this style throughout the manuscript."

Response 4: All citations in the manuscript have been revised to follow the suggested citation style.

Comments 5: "5- Line 24; I would say “patient conditions” instead of “patient outcomes”."

Response 5: This suggestion has been implemented, and "patient outcomes" has been replaced with "patient conditions."

Comments 6: "6- Lines 32-36; use Arabic numbers (1, 2, 3, etc.) for section names to be consistent with the numbering style of the paper sections. Please apply the same in all other instances in the manuscript."

Response 6: The section numbering has been revised, and Arabic numbers have been used throughout the manuscript.

Comments 7: "7- Line 57; what is “SML”?"

Response 7: "SML" stood for "Systematic Mapping of Literature," but the acronym has been removed to avoid confusion.

Comments 8: "8- Line 81; footnotes 2 and 3 should be in this page (not in the next page)."

Response 8: The placement of footnotes has been corrected.

Comments 9: "9- Line 85; language correction: “….., which resulted in the selection of 54 articles”."

Response 9: This language correction has been applied.

Comments 10: "10- Line 85; I suggest adding a sentence at the end of this line: “Further article selection criteria are applied as explained next”."

Response 10: The suggested sentence has been added.

Comments 11: "11- Some entries of table 2 need explanation in the text before the table:
(a) CE-01 and CE-02: how did these articles appear in your search results, while your search string has the keywords “healthcare” and “wearable” as explicit terms?
b) CE-03: what is exactly meant by “duplicate articles”? How did they appear in your results? You need to explain this in the light of the “research repositories” you used.
c) CE-04: do the “Scopus” and “IEEE” repositories include tutorials and courses?"

Response 11: 
(a) An explanation has been added to clarify that articles appeared in the search results because the keywords could be found in titles, abstracts, or keywords, even if they were used as examples or future trends.
(b) The concept of "duplicate articles" has been explained in the text, clarifying how they appeared in the results based on the repositories used.
(c) You are correct. The search criteria have been adjusted, as these repositories do not include tutorials or courses.

Comments 12: "12- You need to describe How the criteria in tables 1 and 2 were implemented? Did you do this manually (i. e. by quick or thorough reading)? What measures did you take in order to maintain objectivity while implementing the criteria?"

Response 12: The criteria were applied manually through a thorough reading process. We have added text to describe the steps taken to ensure objectivity throughout the implementation.

Comments 13: "13- Line 86; how many articles underwent the list of quality questions? Which block in figure 1 represents this list of questions?"

Respose 13: A total of 26 articles were removed during the quality assessment process. The text and Figure 1 have been updated to clarify this and represent the relevant block.

Comments 14: "14- I noticed that some questions can have a “non-applicable” answer. For example, if you have a “theoretical analysis” article as indicated in table 1, then questions related to data collection and sample size will not be applicable. How did you deal with these cases?"

Response 14: In such cases, the non-applicable questions were disregarded, and the remaining questions were scored, maintaining a threshold of 70%. The text has been updated to reflect this approach.

Comments 15: "15- I have a number of comments on figure 1:
a) I don’t see the need for the yellow title bar at the top of the figure.
b) Some information are included in the figure of not included in the body text such as:
— “Reports sought for retrieval” and “reports not retrieved”. What do you mean by these?
— The contents of the box to the right of “reports assessed for eligibility (n=54)”. It is not clear how did you decide upon “independent study design” or “lack of specificity in technology”, etc.
— You basically need to unify the description of your selection/filtering procedures given in the body text with the information given in figure 1.
c) In the last box, why do you refer to studies as being “New”? And what is meant by “reports of a new included studies”?
d) The figure caption includes information that should be in the body text: “After the inclusion ….. by the authors”. Please only keep the first sentence: “Search methodology …. mapping” as your figure caption and move the rest to body text."

Response 15: The figure was entirely changed and the text better written to match each other.

Comments 16: "16- To sum up, the major comment on section 3.2 is that you need to describe, in a clear manner, how did you select 28 articles from an original list of 171 articles? Please do not skip any details or leave any ambiguous term/step."

Response 16: Additional details have been added to Section 3.2 to clearly explain how the final 28 articles were selected from the initial 171.

Comments 17: "17- Section 3.3 title; delete “AND” and use a comma “,” instead. Also, replace “&” by “and”."

Response 17: This has been corrected as suggested.

Comments 18: "18- Line 108; 16 articles or 28 articles?"

Response 18: The text has been corrected to reflect 28 articles.

Comments 19: "19- Page 5; utilise the empty space by splitting table 3 into two parts. The table can be continued in the next page."

Response 19: The formatting of Table 3 has been adjusted to improve spacing, and the table is now split across pages where necessary.

Comments 20: "20- Line 127; “Section 3” instead of “III”."

Response 20: Fixed.

Comments 21: "21- Fingers 2; add axes titles: “Years” and “No. of articles”.

Response 21: Axes titles have been added to Figure 2 as suggested.

Comments 22: "22- Line 130; “Another analysis” instead of “A first analysis”.

Response 22: This has been corrected.

Comments 23: "23- Line 135; correction: “…. blood pressure estimation and others cover remaining selected articles”.

Response 23: The correction has been made.

Comments 24: "24- Figure 3; revise the numbers of studies in the brackets. Did you use the number of the reference or the sequence number from table 3?
For example, [27] refers to “Communicable Disease Detection” in figure 3, but in table 3 the same article is [30]."

Response 24: The numbers in Figure 3 and Table 3 have been updated to ensure consistency.

Comments 25: "25- Figure 3; make sure that all studies are included in the map, even if you have to add a category with the title “others”.

Response 25: All studies are now included in the map

Comments 26: "26- Figure 3; caption is too long and includes a sentence that can be moved to body text.

Response 26: The caption has been shortened, and the additional information has been moved to the body text.

Comments 27: "27- This is a general comment on all tables: in tables 1-3, you used all-capital case letters for the table titles, while for table 4 onwards you used normal sentence case letters. Please unify the font style for all table titles in the paper?"

Response 27: All table titles have been updated to use a consistent style.

Comments 28: "28- For tables 4 and onward; capitalise the first letter in column titles. Example “Dataset” column title in table 6. Also capitalise the first letter of the text in each cell. Example “Self collected data” in table 6."

Response 28: This has been corrected for all relevant tables.

Comments 29: "29- Line 172; try to comment on the meaning and implication of “accuracy” in this context, so that the reader is not misled to an invalid comparison of the different techniques listed in table 6."

Response 29: A detailed explanation regarding the meaning and implications of "accuracy" has been added.

Comments 30: "30- Table 8; F1-scores are not included, but the table title states that they are included.

Response 30: The title has been revised and fixed.

Comments 31: "31- In table 10; please use a comma “,” instead of “+” to separate the different datasets.

Response 31: This correction has been applied.

Comments 32: "32- Line 242; “on the other hand”.

Response 32: Fixed.

Comments 33: "33- Table 12; F1-scores are not included, but the table title states that they are included.

Response 33: The title has been revised and fixed.

Comments 34: "34- In table 13; please use a comma “,” instead of “+” to separate the different terms.

Response 34: This correction has been made.

Comments 35: "35- Line 328; I think you mean “As shown in section 4”.

Response 35: The text has been revised as suggested.

Comments 36: "36- Line 343; same as previous comment.

Response 36: The correction has been applied.

Comments 37: "37- Lines 343-348; I don’t see any comments/observations on the “Machine learning algorithms” as part of RQ3.

Response 37: Additional insights regarding machine learning algorithms have been added to RQ3 to improve clarity.

Comments 38: "38- Footnote 4 at the bottom of the page seems to talk about ref [9], but inside table 18 the footnote mark is placed next to ref [8]. Make necessary correction.

Response 38: The reference and footnote have been corrected.

Comments 39: "39- Conclusion section; I expect the authors to expand their conclusions to include more critique findings such as:
— What challenges have you found in this field?
— Where do you see the future research trends headed?
— What could make a breakthrough in the field?
…. and so on."

Response 39: The Conclusion section has been expanded to include discussions on current challenges, future trends, and potential breakthroughs in the field.

Reviewer 3 Report

Comments and Suggestions for Authors

The paper presents a systematic mapping of literature on the application of Machine Learning on edge computing and wearable devices for healthcare. With several adjustments based on the suggestions provided, this paper has the potential to make a strong impact in the emerging field.

  • The authors specified that the studies included in the review are 28 as mentioned in figure 1. In section 3.3 just after the figure, the authors are mentioning 16 articles, in addition the references are 33. The authors are invited to clarify this discrepancy.
  • The authors are invited to revise the form and the figure sizes to avoid having almost blank pages
  • In table 3, the authors selected the papers used for the literature mapping, The references 1, 2, 3, 32 and 33 are not included. The authors are invited to clarify this discrepancy.
  • In the presentation of figure 2, the authors mentioned that: “all selected articles are from the last six years, with the majority published within the last three years”. In figure 2, the selected articles are from 2018 and 2024 that make it the last 7 years instead of 6. In addition, the majority is in the last 4 years from 2021 till 2024 and not last three years. The authors are invited to correct this part.
  • In figure 3, a conceptual mapping of healthcare applications is presented. The references 8, 18, 22, 30, 32 and 33 are not included. The reference 27 is not for communicable disease detection as per table 3, it should be reference 30. The prediction of heart disease is not included. The authors are invited to revisit the figure and correct it to cover all the 33 papers.
  • The authors are invited to correct section 4.6 as several references are wrong and does not reflect table 3 information. Reference 9 should be 8 and 8 should be 9, 21 should be 11…
  • In section 5, the authors are based on the section 4, so all section 5 should be corrected to section 4.
  • The authors are invited, and for more clarity and consistency, to include more metrics than accuracy to cover all the applications of section 4.6 since they present a significant part of the selected article for mapping.
  • The discussion on the ML algorithms used could be expanded. While neural networks are frequently mentioned, deeper insights into why specific models are used in certain applications, and how they perform comparatively, would enhance the value of the review.
  • In summarizing the findings, the authors are invited to add additional visualizations, such as diagrams illustrating how ML algorithms integrate with specific healthcare devices, could improve comprehension

Author Response

Comments 1: "The authors specified that the studies included in the review are 28 as mentioned in figure 1. In section 3.3 just after the figure, the authors are mentioning 16 articles, in addition the references are 33. The authors are invited to clarify this discrepancy."

Respose 1: Thank you for your observation. We have corrected the inconsistency in Section 3.3. Regarding the number of references, we would like to clarify that, in addition to the 28 articles included in the systematic review, additional references were cited to provide context and enhance the overall quality of the text. These include foundational papers on the methodological guidelines for conducting systematic reviews, as well as references to hardware specifications used in some of the studies analyzed. This explains the discrepancy between the number of reviewed articles and the total number of references cited.

Comments 2: "The authors are invited to revise the form and the figure sizes to avoid having almost blank pages"

Response 2: The figures and tables were fixed to better fit on the pages and avoid blank spaces.

Comments 3: "In table 3, the authors selected the papers used for the literature mapping, The references 1, 2, 3, 32 and 33 are not included. The authors are invited to clarify this discrepancy."

Response 3: Thank you for your comment. These articles were included to enhance the contextual framework of the study rather than being part of the literature mapping itself. Some of them discuss foundational aspects of the research topic and provide the motivation for this study, while others offer technical details regarding the hardware used in some of the selected studies. As such, they were not included in Table 3, which lists only the papers directly analyzed for the literature mapping.

Comment 4: "In the presentation of figure 2, the authors mentioned that: 'all selected articles are from the last six years, with the majority published within the last three years.' In figure 2, the selected articles are from 2018 to 2024, which makes it the last seven years instead of six. In addition, the majority is from the last four years (2021 to 2024) and not the last three years. The authors are invited to correct this part."

Response 4:
Thank you for pointing out this discrepancy. We have made the necessary corrections to accurately reflect the publication years, specifying that the selected articles cover the last seven years, with the majority published in the last four years.

Comment 5: "In figure 3, a conceptual mapping of healthcare applications is presented. The references 8, 18, 22, 30, 32, and 33 are not included. Reference 27 is not for communicable disease detection as per table 3; it should be reference 30. The prediction of heart disease is not included. The authors are invited to revisit the figure and correct it to cover all the 33 papers."

Response 5: We have revisited the figure and made the necessary corrections to ensure that it accurately represents all 33 papers. These adjustments include the proper classification of references and the inclusion of heart disease prediction.

Comment 6: "The authors are invited to correct section 4.6 as several references are wrong and do not reflect table 3 information. Reference 9 should be 8, and 8 should be 9, 21 should be 11…"

Response 6: The errors in section 4.6 have been corrected, and the references now align with the information in Table 3.

Comment 7: "In section 5, the authors are based on section 4, so all of section 5 should be corrected to reference section 4."

Response 7: We have made the necessary corrections, ensuring that section 5 correctly references section 4.

Comment 8: "The authors are invited, for more clarity and consistency, to include more metrics than accuracy to cover all the applications of section 4.6 since they present a significant part of the selected articles for mapping."

Response 8: We have included additional performance metrics beyond accuracy to cover the range of applications discussed in section 4.6, along with a disclaimer explaining the context and relevance of these metrics.

Comment 9: "The discussion on the ML algorithms used could be expanded. While neural networks are frequently mentioned, deeper insights into why specific models are used in certain applications, and how they perform comparatively, would enhance the value of the review."

Response 9: We have expanded the discussion to provide deeper insights into the rationale behind the selection of specific machine learning models in various applications, as well as a comparative analysis of their performance across different use cases.

Comment 10: "In summarizing the findings, the authors are invited to add additional visualizations, such as diagrams illustrating how ML algorithms integrate with specific healthcare devices, to improve comprehension."

Response 10: We have included a new diagram to visually illustrate how machine learning algorithms integrate with specific healthcare devices, providing enhanced clarity and comprehension for the reader.

Round 2

Reviewer 1 Report

Comments and Suggestions for Authors

The authors have modified the manuscript in a satisfactory manner. It’s acceptable for publication in the current form.

Author Response

Thank you for your positive feedback and for the time and effort spent reviewing our manuscript. We are grateful for your constructive comments, which have helped improve the quality of our work. We are pleased to hear that the revised version is now acceptable for publication.

Reviewer 3 Report

Comments and Suggestions for Authors

The paper presents a systematic mapping of literature on the application of Machine Learning on edge computing and wearable devices for healthcare. After authors adjustments, we think that this paper still missing the following:

  • In figure 3, a conceptual mapping of healthcare applications is presented. The references 32 till 34 are not included. In addition, references 4, 5 and 6 are still existent in the figure while they are not in table 3. Reference 30 should be fall detection as in table 3 while it is mentioned as Communicable disease detection in figure 3. The authors are invited to correct the figure again.
  • In section 4 and all its subsection, the references are not the same as in the figure 3, while it is the results section. Section 4.1 has different references than presented in figure 3, same for section 4.2, and so on

Author Response

Comment 1: "In figure 3, a conceptual mapping of healthcare applications is presented. The references 32 till 34 are not included. In addition, references 4, 5 and 6 are still existent in the figure while they are not in table 3. Reference 30 should be fall detection as in table 3 while it is mentioned as Communicable disease detection in figure 3. The authors are invited to correct the figure again.
In section 4 and all its subsection, the references are not the same as in the figure 3, while it is the results section. Section 4.1 has different references than presented in figure 3, same for section 4.2, and so on"

Respose 1: Thank you for your valuable feedback. We acknowledge the inconsistencies in Figure 3 and Section 4, as you have pointed out. We have carefully reviewed and corrected these issues.